# Native flagellar MS ring is formed by 34 subunits with 23-fold and 11-fold subsymmetries

Akihiro Kawamoto [1,2,7], Tomoko Miyata [1,7], Fumiaki Makino [1,3], Miki Kinoshita [1], Tohru Minamino [1], Katsumi Imada [4], Takayuki Kato [1,2 ✉] & Keiichi Namba [1,5,6 ✉]

The bacterial flagellar MS ring is a transmembrane complex acting as the core of the flagellar motor and template for flagellar assembly. The C ring attached to the MS ring is involved in torque generation and rotation switch, and a large symmetry mismatch between these two rings has been a long puzzle, especially with respect to their role in motor function. Here, using cryoEM structural analysis of the flagellar basal body and the MS ring formed by full-length FliF from *Salmonella enterica*, we show that the native MS ring is formed by 34 FliF subunits with no symmetry variation. Symmetry analysis of the C ring shows a variation with a peak at 34-fold, suggesting flexibility in C ring assembly. Finally, our data also indicate that FliF subunits assume two different conformations, contributing differentially to the inner and middle parts of the M ring and thus resulting in 23- and 11-fold subsymmetries in the inner and middle M ring, respectively. The internal core of the M ring, formed by 23 subunits, forms a hole of the right size to accommodate the protein export gate.

[1] Graduate School of Frontier Biosciences, Osaka University, Suita, Osaka, Japan. [2] Institute for Protein Research, Osaka University, Suita, Osaka, Japan. [3] JEOL Ltd., Akishima, Tokyo, Japan. [4] Department of Macromolecular Science, Graduate School of Science, Osaka University, Toyonaka, Osaka, Japan. [5] RIKEN Center for Biosystems Dynamics Research and SPring-8 Center, Suita, Osaka, Japan. [6] JEOL YOKOGUSHI Research Alliance Laboratories, Osaka University, Suita, Osaka, Japan. [7] These authors contributed equally: Akihiro Kawamoto, Tomoko Miyata. ✉email: tkato@protein.osaka-u.ac.jp; keiichi@fbs.osaka-u.ac.jp

Bacteria actively swim in liquid environments by rotating long, helical filamentous organelle called the flagellum. The bacterial flagellum is supramolecular motility machinery consisting of the basal body that acts as a bi-directional rotary motor, the filament that functions as a helical propeller, and the hook as a universal joint connecting the basal body and the filament to transmit motor torque to the filament[1–3]. The basal body consists of the MS ring, C ring, LP ring, and the rod (Fig. 1a). The MS ring is a transmembrane protein complex made of FliF and is the base for flagellar structure, assembly and function. The MS ring is not only the mounting platform for the C ring, which is the cytoplasmic part of the basal body formed by the switch proteins FliG, FliM and FliN and acts as a bi-directional rotor of the flagellar motor as well as the switch regulator of the rotation direction, but is also a housing for the type III protein export apparatus that exports flagellar axial proteins for their assembly at the distal growing end of the flagellum[2–4]. The rod is a drive shaft of the motor, transmitting motor torque to the hook and filament and is a helical assembly of four different rod proteins, FlgB, FlgC, FlgF and FlgG, attached to the MS ring and export gate through an adaptor protein, FliE. The LP ring is a bushing tightly embedded in the peptidoglycan layer as well as the outer membrane and surrounds the distal, thicker tubular part of the rod formed mainly by FlgG[5,6] to stabilize the high speed rotation of the motor[1–3]. Motor torque is generated by cyclic interactions and dissociation of FliG at the top end of the C ring with the cytoplasmic domain of the transmembrane stator complex formed by MotA and MotB, which also acts as a proton channel to couple proton influx across the membrane with torque generation[7], where multiple number of stator units surround the rotor to become active depending on the external load[8].

For a long time until recently, the MS ring had been believed to have 26-fold rotational symmetry whereas the C ring has 34-fold symmetry with some minor variations based on biochemical and

structural studies[9–13]. This symmetry mismatch between the two rotor complexes had been a focus of debates on its possible role in the C ring assembly, torque generation and stepping rotation and the basis for the mechanistic understanding of torque generation mechanism[11,14,15]. This puzzle seemed resolved by the recent high-resolution structure of the MS ring with 33-fold symmetry with a variation from 32 to 35 (ref. [16]) because the C ring also shows a similar symmetry variation peaked around 34- or 35-fold[12,17]. However, it still remained ambiguous whether their symmetries are actually matched in the native motor structure, because the distributions of the symmetry variations are clearly different between the two rings, with a peak around 34- or 35-fold for the C ring[12,17] and 33-fold for the MS ring[16]. Also, the MS ring structures were analyzed with those formed by overexpressed FliF[16], possibly with some C-terminal truncations as suggested in the SDS-PAGE band pattern of the FliF protein preparation, and this may have produced the observed symmetry variation.

There is also another interesting symmetry mismatch between the MS ring and the rod, which assembles directly on the MS ring while having a helical tubular structure with 11 protofilaments, with a helical symmetry of about 5.5 subunits per one turn of helix. The core structure of the type III protein export gate composed of FliP, FliQ and FliR[18] is thought to be located inside the MS ring with a helical nature in their assembly to be directly connected with the proximal part of the rod formed by FlgB, FlgC and FlgF via an adapter protein FliE[19]. However, it remains unknown how the symmetry mismatch between the protein export gate-rod and the MS ring is resolved to make flagellar assembly possible by the MS ring as the assembly template.

Here, we report electron cryomiscroscopy (cryoEM) structural analyses of the flagellar basal body and the MS ring formed by full-length FliF from *Salmonella enterica* serovar Typhimurium (hereafter referred to *Salmonella*) and that the symmetry of the native MS ring is 34-fold with no variation. Our own symmetry analysis of the C ring of the basal body, which is more precise

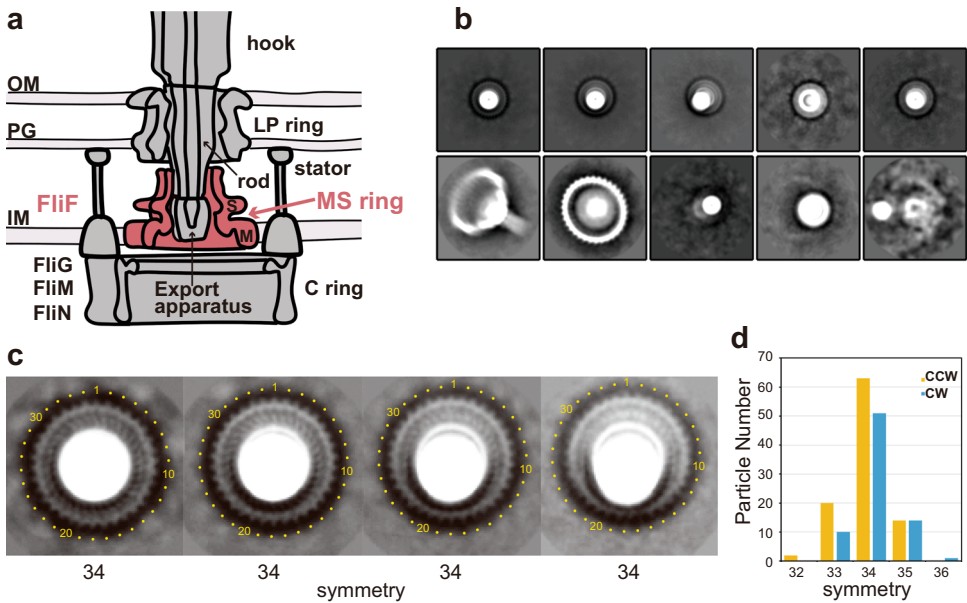

**Fig. 1 Schematic diagram and cryoEM image analysis of the flagellar basal body. a** Schematic diagram of the flagellar basal body. IM: inner membrane, PG: peptidoglycan layer, OM: outer membrane. **b** Representative 2D class average images in the first round. The upper row images show end-on views of the MS ring with rod called rivet (see Supplementary Fig. 1a) except for the second from the right; in the lower row, the two panels on the left show the rivet with the C ring attached. **c** Images showing the rotational symmetry of the S ring are extracted from the second round of 2D class average of the rivet class images and magnified. The rotational symmetries obtained by image analysis as shown in Supplementary Fig. 1 are indicated below, presenting direct evidence for the 34-fold symmetry of the native MS ring. **d** The rotational symmetry distributions of the C ring: CCW in yellow is the wild-type motor in the CCW state (98 particles in total); and CW in blue is the CW-locked motor by FliG ΔPAA mutation (75 particles in total).

than the previous one[12], shows a variation with a peak at 34-fold in agreement with the previous observation[12], indicating that there are small symmetry mismatches between the two rings possibly generated by some flexibility in the initiation of C ring assembly by FliG around the MS ring. The MS ring structure also shows 23-fold and 11-fold symmetries in the inner and middle parts of the M ring, respectively, possibly to accommodate the type III protein export gate to assemble at the center of the MS ring. The structure of the S ring and cylindrical collar in the upper part of the MS ring is resolved at high resolution, and mutational analyses show that conserved residues of FliF are responsible not only for MS ring formation by stabilizing intersubunit interactions but also for the assembly of the type III protein export apparatus and the rod within the MS ring. Thus, it is now clear that the MS ring and C ring are tightly bound to each other to act as a rotor unit of the flagellar motor to generate and transmit torque to the rod, hook and filament for bacterial motility.

## Results

**CryoEM structural analysis of the basal body.** We isolated the flagellar basal body from a *Salmonella* mutant strain HK1003 [*flgEΔ(9-20) ΔclpP*::Cm deletion mutant], in which the number of the basal body was increased by the deletion of the ClpP protease[20], and analyzed the structure by cryoEM image analysis. Most of the isolated basal bodies are missing the LP ring because the LP ring easily dissociates from the rod in the absence of the hook (Fig. 1b, Supplementary Fig. 1a). This structure is called a rivet because of its shape, and this partial basal body structure is advantageous for visualizing the symmetry of the MS ring in the end-on view because the LP ring normally above the MS ring disturbs the visualization of the MS ring image. The two-dimensional (2D) class average images of its end-on views thus allowed the determination of the MS ring symmetry (Fig. 1c, Supplementary Fig. 1b). Of the 34,896 particle images of the basal body that were picked up from the 1578 cryoEM micrographs, 23,104 particle images were near end-on views, separated into four classes with slight differences in the orientation, where each of the four class images clearly showed the 34-fold rotational symmetry (Fig. 1c). So, the MS ring structure in the native flagellar basal body is composed of 34 FliF subunits.

We also analyzed the symmetry of the C ring of the native flagellar basal body. Although the symmetry of the C ring in the basal body has already been studied by cryoEM image analysis, the previous study used an indirect method in which side view images of the C ring were subjected to multireference alignment against projections of three-dimensional (3D) models with symmetries from 32- to 36-fold to sort the particle images into different rotational symmetries[12], and this may have produced some errors by misalignment and possible deformation of the C ring from a perfect cylinder. We therefore used only end-on views of the C ring that show blobs of subunits and analyzed the rotational symmetry (Supplementary Fig. 1c). Since the flagellar motor can switch the rotation between the counter-clockwise (CCW) and clockwise (CW) direction, we looked at both CCW and CW locked C ring structures. As shown in Fig. 1d, the C ring symmetry has a certain range of variation from 32- to 35-fold for CCW and 33- to 36-fold for CW, but about two-thirds are 34-fold symmetry with rather minor populations of non 34-fold symmetries (CCW: 32-fold, 2%; 33-fold, 20%; 34-fold, 66%; 35-fold, 13%, 36-fold, 0%; CW: 32-fold, 0%; 33-fold, 13%; 34-fold, 68%; 35-fold, 20%; 36-fold, 1%). This is in marked contrast to much higher populations of non 34-fold symmetries (32-fold, 2%; 33-fold, 17%; 34-fold, 42%; 35-fold, 30%; 36-fold, 7%) estimated in the previous study[12]. Thus, although the symmetry mismatch between the MS ring and C ring is present, the probabilities of C

ring assembly around the MS ring with extra or less FliG subunits to the FliF subunits is small, and the number of extra or less FliG subunits is mostly up to one, indicating that the flexibility in the template driven C ring assembly by FliG around the 34 FliF-subunit MS ring that causes a symmetry mismatch between these two rings is not so high.

**CryoEM structural analysis of the MS ring formed by over-expressed FliF.** We overexpressed full-length *Salmonella* FliF in *E. coli* and isolated the MS ring from the membrane fraction by sucrose gradient purification (Supplementary Fig. 2a, b). Although the ring-shaped particles were clearly observed in negative stained EM images (Supplementary Fig. 2c), it was difficult to determinate the structure of the MS ring by cryoEM image analysis because most of the particles were attached to the edge of carbon holes. To alleviate such awkward characteristics of the MS ring, we optimized the purification procedure using LMNG as a detergent, which drastically improved the particle dispersion and density in the holes and allowed high quality image data collection in sufficient particle numbers. A total of 339,861 particles were extracted from 1589 micrographs and were analyzed. Representative 2D class averages showed homogeneous ring particles with clear 34-fold symmetry (Fig. 2a, Supplementary Figs. 3 and 4). We did not observe any other symmetries in the 2D class averages, indicating that the MS ring formed by full length FliF consists of 34 subunits and shows no variation in the ring stoichiometry, just as we observed for the MS ring in the native basal body structure as described above.

We then carried out 3D image reconstruction of the MS ring, but the resolution did not extend beyond 10 Å without rotational

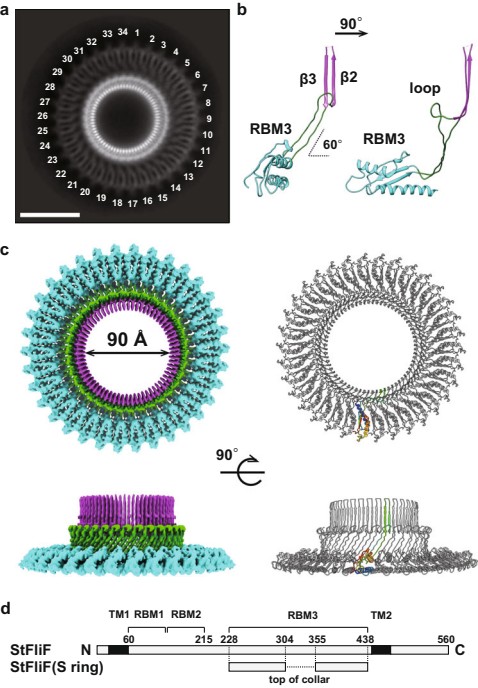

**Fig. 2 Structure of the S ring and collar of the MS ring formed by full-length FliF. a** A representative 2D class average image showing 34-fold symmetry. **b** Cα ribbon representation of the RBM3 domain model forming the S ring and collar in two orthogonal side views. **c** CryoEM 3D image reconstruction of the S ring and collar (left) and Cα ribbon diagram of the atomic model (right) in end-on (upper) and side (lower) views. **d** Domain organization of *Salmonella* FliF, indicating the region forming the S ring and collar. TM1 and TM2: transmembrane regions; numbers above the bars: residue numbers in the FliF sequence.

symmetry. We therefore performed iterative 3D refinement with 34-fold rotational symmetry. This process dramatically improved the resolution of the 3D map to 3.7 Å resolution (Supplementary Fig. 3) (EMDB ID: EMD-30612) and enabled us to construct the atomic model of the S ring and collar (Fig. 2b, c) (PDB ID: 7D84). Unfortunately, however, a large part of the M ring was somehow disordered in this particular construct.

The S-ring and collar is formed by residues 228–438 of *Salmonella* FliF (Fig. 2d), which correspond to the latter half of the periplasmic region. The monomeric structure consists of two structural regions: a globular domain with αββαβ motif that is known as a ring building motif (RBM) (residues 228–270 and 382–438 designated as RBM3 in Fig. 2b); and a long, extended up-and-down β structure (residues 271–381) consisting of a set of antiparallel chains and a set of antiparallel β strands, β3 and β4, forming a β hairpin with invisible 51 residues possibly forming a flexible loop (Fig. 2b). The chain connecting β3 and β4 at the top of the collar (residues 305–354) was not modeled due to the poor density (Fig. 2b–d). The S ring is made up of 34 RBM3 domains horizontally packed with their major axis oriented in the radial direction to form a ring with a diameter of 24 nm. The long β hairpins in the upper part of the collar are vertically lined up to form a 68-stranded cylindrical β-barrel structure. The two extended antiparallel chains connecting the RBM3 domain and the vertical β hairpin are inclined about 30° from the ring axis. The model was nearly identical to the corresponding part of the recent MS ring structure with 34-fold symmetry[16].

The overall structure of the S-ring and collar resembles the SpoIIIAG structure[21] despite the low sequence identity (<15%) (Supplementary Fig. 5a, b). The unique large β-barrel structure composed of vertically arranged β-strands is present in the SpoIIIAG ring. The ring arrangement of the RBM3 domains is also similar to that of the SpoIIIAG, although the RBM domain of the SpoIIIAG ring is tilted about 14° compared with the nearly horizontal orientation of that of the S ring (Supplementary Fig. 5a, b). The ring formation through the RBM domain is commonly found in the T3SS injectisome, and the domain arrangement of the S ring is similar to those in the T3SS ring except for the D2 domain of PrgH, which uses completely different surfaces for ring formation[22–26].

The chain folding arrangement of FliF in the collar is also similar to those seen in the secretin rings of the type III secretion system (T3SS) of bacterial pathogens[27]. Unlike them, however, only a small number of hydrogen bonds are formed between these two chains in the flagellar MS ring because the two antiparallel chains are twisted and apart from each other toward the bottom of the cylindrical β-barrel structure, even though the phi-psi angles of each residue in these extend chain are mainly those of the β type conformation. Residues 283–293 connecting to β3 of the vertical β hairpin is looping out to form a structure that looks like a saucer to the cup upon assembly into the ring. The two antiparallel chains are swapped right and left before connecting to the vertical β hairpin in the upper part of the collar (Supplementary Fig. 5c).

**Mutation analysis of FliF for the MS ring formation**. We examined whether the atomic model of the S ring represents the structure of the physiologically functional MS ring of the flagellar basal body by mutations of FliF followed by assays of cell motility, flagellar protein export and assembly of the basal body and MS ring. The subunit interface of the RBM3 domains in the S ring is mediated by both the polar and hydrophobic interactions. Ile-252, Leu-253, and Val-266 are relatively well conserved between FliF and SpoIIIAG[21] (Fig. 3a). Ile-107 and Val-120 of SpoIIIAG corresponds to Ile-252 and Val-266 of FliF, respectively, and the

replacement of each residue by Arg inhibits SpoIIIAG ring formation. We therefore replaced Ile-252, Leu-253, and Val-266 by Ala or Arg and analyzed the effects of these mutations on cell motility in soft agar to test whether these three residues are involved in MS ring formation (Fig. 3b). These substitutions did not significantly affect the steady cellular level of FliF as judged by immunoblotting with polyclonal anti-FliF antibody (Fig. 3c). Wild-type FliF fully restored motility of a Δ*fliF* mutant. The V266A mutant variant complemented the Δ*fliF* mutant to the wild-type level, and the L253A mutant variant restored the motility to a considerable degree, but the I252A mutant variant did so only poorly. The I252R, L253R, and V266R mutant variants did not complement the Δ*fliF* mutant at all (Fig. 3b). In agreement with this, the I252A and L253A mutants formed a few flagellar filaments but the I252R, L253R, and V266R mutants did not at all (Fig. 3d).

The MS ring is also a housing for the flagellar type III protein export apparatus[28,29]. Therefore, we tested whether the poor flagellar formation by the *fliF*(I252A) and *fliF*(L253A) mutant strains compared with the wild-type and *fliF*(V266A) mutant strains is a consequence of their reduced protein export activity. The cytoplasmic face of the MS ring is the template for assembly of the C ring by the switch proteins FliG, FliM, and FliN[2–4], and the C ring is also a housing of the cytoplasmic part of the type III protein export apparatus. The loss of C ring considerably reduces the flagellar protein export activity and thereby impairs the cell motility[28], we therefore tested whether the I252A and L253A mutations of FliF also affect C ring formation. If the C ring is formed normally on the cytoplasmic face of the MS ring, FliG, FliM and FliN should be detected in the membrane fraction of the cells because the MS ring is a transmembrane complex. No FliF and only a very small amount of FliN were found in the membrane fraction of the Δ*fliF* mutant as expected, but large amounts of FliF and FliN were detected in the membrane fractions of all the *fliF* mutant cells just as observed for wild-type cells (Fig. 3c). This indicates that these FliF mutations do not affect MS-C ring formation at all.

We then examined the secretion levels of the hook-capping protein FlgD, the hook protein FlgE and the filament protein FliC as representative export substrates and found that the reduction in their secretion levels was well correlated with the reduction in the levels of flagellar formation (Fig. 3d, e). The *fliF*(I252A) and *fliF*(L253A) mutant strains produced hook-basal bodies (Fig. 3d) but at a significantly lower level than the wild-type. These results suggest that the I252A and L253A mutations have direct effects on the assembly of the type III protein export apparatus into the MS ring. The I252R, L253R, and V266R mutations abolished flagellar protein export (Fig. 3e), explaining the complete loss of motility of these mutant strains (Fig. 3b).

To examine the effect of these FliF mutations on the stability of the MS ring, we purified the MS rings from *E. coli* cells overexpressing these mutant variants of FliF and observed them by negative staining EM. All the FliF mutations significantly affected the stability, and most of the ring-shaped particles of the size of the MS ring were not completely closed (Fig. 3f). Ile-252 is located at the interface between FliF subunits whereas Leu-253 and Val-266 are not, suggesting that Ile-252 is responsible for intermolecular interactions between FliF subunits whereas Leu-253 and Val-266 are involved in proper folding of each FliF-subunit to form the S ring. To further confirm this, we carried out in vitro disulfide crosslinking experiments.

Since His-263 and Ala-388 are in close proximity to Ile-252 in the atomic model of the S ring (Fig. 3a), we replaced Ile-252, His-263 and Ala-388 by Cys and analyzed cell motility in soft agar plates. Cys mutation of each residue did not drastically affect cell motility. The H263C variant complemented the Δ*fliF* mutant to

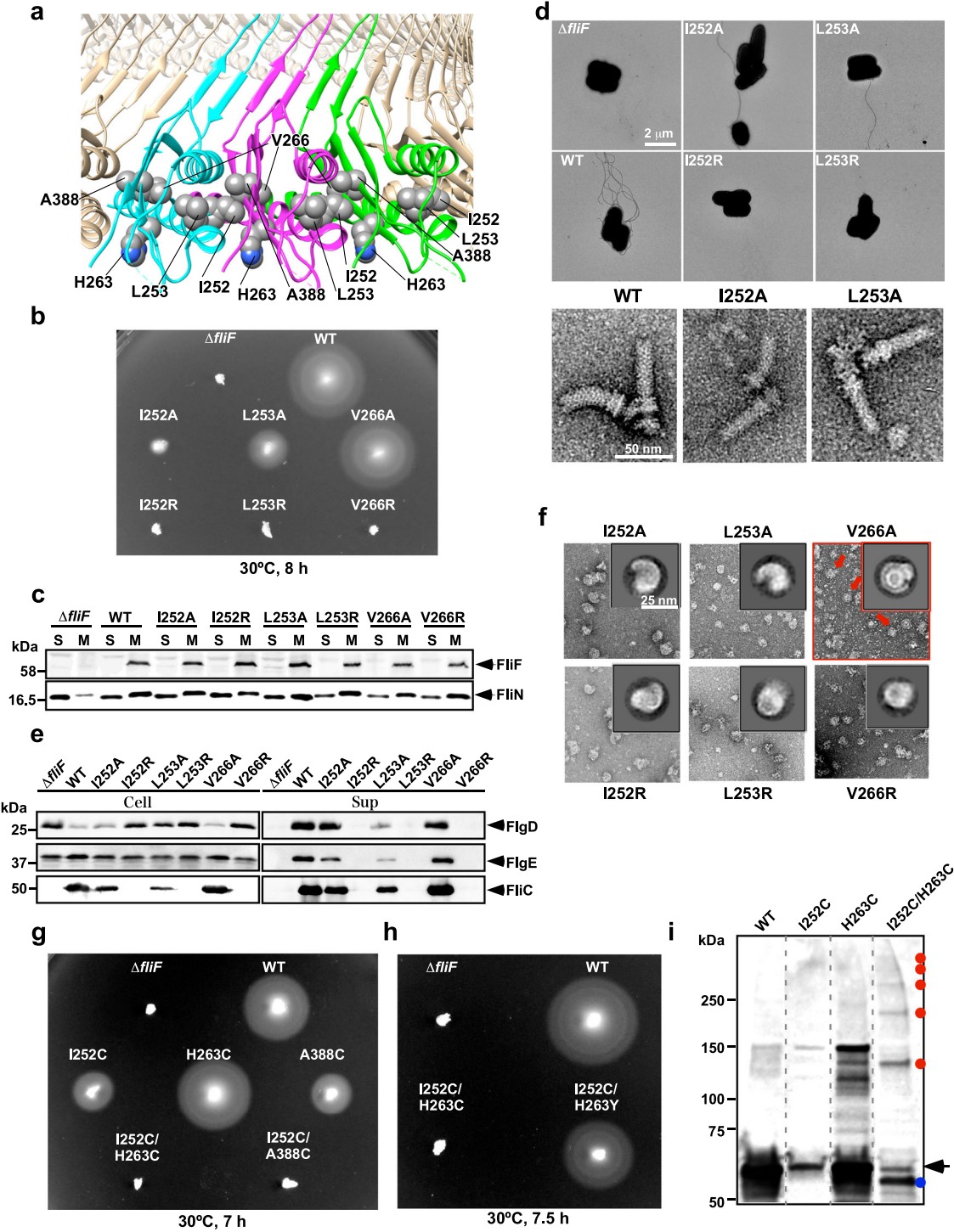

the wild-type level, and the I252C and A388C variants restored the motility to a considerable degree although not to the wild-type level (Fig. 3g). However, neither I252C/H263C nor I252C/ A388C double mutation variant complemented the ΔfliF mutant to restore cell motility, suggesting that crosslinked FliF subunits in the MS ring may have impaired its function, such as insertion of the type III protein export gate into the MS ring. Intersubunit disulfide crosslinking of FliF by the double Cys mutations was confirmed by higher-order oligomeric species observed in the membrane fraction isolated from the fliF(I252C/H263C) mutant upon inducing disulfide crosslinking by adding iodine (Fig. 3i, red dots), confirming that the atomic model of the S ring we built

here is a physiological one. Such oligomers were observed neither for the wild-type, I252C nor H263C mutant (Fig. 3i). The appearance of an extra band of FliF monomer with a slightly faster mobility in the SDS-PAGE gel (Fig. 3i, blue dot) can be due to intramolecular disulfide crosslink, suggesting a significantly different conformation of FliF monomer before its assembly into the MS ring. It is also possible that the conformational change of FliF monomer by intramolecular disulfide crosslink disturbs the proper assembly of the MS ring. The replacement of Cys-263 by Tyr in this double Cys FliF mutant considerably restored the motility (Fig. 3h), suggesting a requirement of flexibility in the MS ring structure for its proper function.

**Fig. 3 Mutational analysis of conserved Ile-252, Leu-253 and Val-266. a** Molecular interface between FliF subunits. Ile-252, Leu-253, and Val-266 are located at an interface between FliF subunits. His-263 and Ala-388 are in close proximity to Ile-252. **b** Motility assay on soft agar of *Salmonella* TH12415 cells harboring a plasmid expressing a FliF mutant. **c** Membrane localization of the MS ring protein FliF and the C ring protein FliN. The membrane fractions of the above transformants were prepared after sonication and ultracentrifugation, subjected to SDS-PAGE and analyzed by immunoblotting with polyclonal anti-FliF or anti-FliN antibody. **d** Electron micrographs of TH12415 cells harboring a plasmid expressing a FliF mutant, and purified hook-basal bodies isolated from WT, I252A and L253A cells. **e** Secretion assays. Whole cell proteins (Cell) and culture supernatant fractions (Sup) were prepared from the above strains. An 8 µl solution of each protein sample, which was normalized to an optical density of $OD_{600}$, was subjected to SDS-PAGE, followed by immunoblotting with polyclonal anti-FlgD (first row), anti-FlgE (second row) or anti-FliC (third row) antibody. **f** Negative-stain EM images of the MS rings isolated from the six FliF mutants. **g** Motility assay of *Salmonella* TH12415 cells harboring a plasmid expressing a FliF mutant. **h** Isolation of pseudorevertants from the I252C/H263C mutant cells. Motility assay of *Salmonella* TH12415 cells harboring a plasmid expressing a FliF mutant. **i** Disulfide crosslinking of FliF–FliF interface. The membrane fractions were prepared from TH12415 cells expressing wild-type FliF, FliF(I252C), FliF(H263C) or FliF (I252C/H263C), and disulfide crosslinking were induced by iodine. Each sample was normalized to an optical density of $OD_{600}$, treated with N-ethylmaleimide, followed by non-reducing SDS-PAGE with a 4–15% gradient SDS-gel and finally by immunoblotting with polyclonal anti-FliF antibody. Oligomeric and monomeric forms are indicated by red dots and an arrow, respectively. A blue dot presumably indicates FliF monomers with an intramolecular disulfide bond. Molecular mass markers (kDa) are shown on the left and the original immunoblots are shown in Source data. The plasmid used in the experiments are listed in Supplementary Table 1. Δ*fliF* and WT are the negative and positive control.

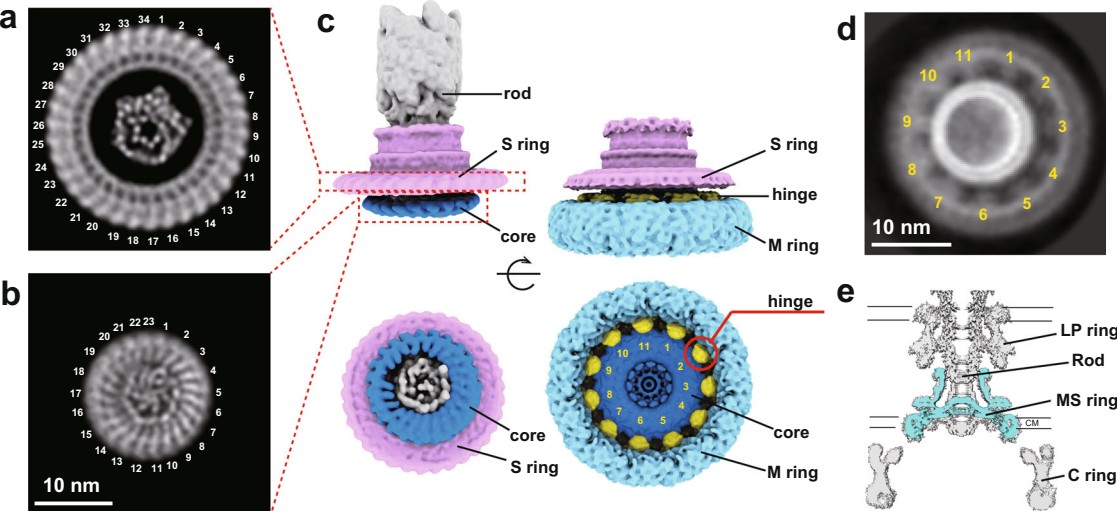

**Fig. 4 CryoEM structure of the 34-subunits MS ring with 23-fold and 11-fold subsymmetries. a**, **b** End-on views of the S ring (**a**) and inner part of the M ring (**b**) of the basal body, showing the 34-fold and 23-fold rotational symmetries, respectively. The dot- and rod-like densities at the center of the rings represent the structure of the type III protein export gate formed by FliP, FliQ, and FliR. **c** 3D reconstructions of the basal body (left) and the MS ring formed by full-length FliF (right) in side (upper) and end-on (lower) views. **d** 2D class average image of the MS ring in end-on view, revealing the 11-fold rotational symmetry in the M ring. **e** A central section of the MS ring density through the axis (blue) is well superimposed on that of the basal body density, indicating the structural identity of the MS ring formed by full-length FliF with that of the basal body.

**Internal structure of the MS ring with 23-fold and 11-fold subsymmetries.** The 3D reconstruction of the basal body at about 6.8 Å resolution (Fig. 4a–c, Supplementary Fig. 6) (EMDB ID: EMD-30613) from cryoEM image data shown in Fig. 1b revealed interesting features in the inner core of the M ring in addition to the 34-fold rotational symmetry of the S ring (Fig. 1c). The inner part of the M ring is a flat ring with clear 23-fold symmetry, and its central hole accommodates a helical assembly of many rod-shaped densities possibly representing α-helices, which looks very similar to the structure of the export gate complex formed by FliP, FliQ, and FliR[19]. As we stated earlier, cryoEM 3D reconstruction of the MS ring structure from an FliF overexpression construct had a large part of the M ring density disordered and therefore not showing its internal feature in detail (Supplementary Fig. 3), but the MS ring formed by FliF overexpressed from another construct produced the entire MS ring density that can be identified in the basal body (Fig. 4c, e, Supplementary Fig. 7) (EMDB ID: EMD-30363). We identified an 11-fold symmetry density features just outside the inner M ring with 23-fold symmetry. Even 2D class average images showed the 11-fold symmetry just outside the collar (Fig. 4d, Supplementary Fig. 7). We

therefore enforced 11-fold symmetry on the entire MS ring structure to see which parts of the MS ring have 11-fold symmetry (Fig. 4c) (EMDB ID: EMD-30361). The outer part of the M ring was featureless but the middle part just outside the inner M ring with 23-fold symmetry showed a strong 11-fold symmetry feature as a hinge connecting to the outer M ring. These structural features indicate that the MS ring is formed by 34 FliF subunits with two distinct conformations. All the 34 subunits contribute their RBM3 in the latter half of the periplasmic region of FliF to the S ring and collar, but 23 copies form the inner M ring and 11 copies form the hinge with their RBM1 and RBM2 in the former half of the periplasmic region (Figs. 2 and 4c).

CryoEM structural analysis of the export gate complex formed by FliP, FliQ, and FliR overexpressed in *E. coli* has revealed a helical nature of multi-subunit assembly of the $FliP_5FliQ_4FliR$ (10 subunits) and $FliP_5FliQ_4FliR_1FlhB_1$ (11 subunits) complexes[19,30], and the helical parameter is similar to those of the flagellar axial structures, such as the rod, hook and filament, with about 5.5 subunits per turn of the 1-start helix, which can also be regarded as a tubular structure with 11 protofilaments. However, the 11-fold symmetry feature of the internal part of the

M ring is not the one directly associated with the export gate at the center of the M ring. It is the internal core ring of the M ring formed by 23 subunits of RBM2 that forms a hole of the right size with a smooth inner surface to accommodate the export gate at the center of the M ring over a symmetry mismatch.

## Discussion

The long-term puzzle on the role of symmetry mismatch between the MS ring with 26-fold rotational symmetry and the C ring with 34-fold symmetry[9–13] had produced much debate on its role in the C ring assembly, torque generation and stepping rotation[11,14,15]. This puzzle appeared to have been resolved by the recent high-resolution structure of the MS ring with 33-fold symmetry with a variation from 32- to 35-fold[16] because the C ring also shows a similar symmetry variation around 34- or 35-fold[12,17]. However, it was still not clear whether the symmetries of these two rings are actually matched in the native motor structure, because the distributions of the symmetry variations are clearly different between the two rings, with a peak around 34- or 35-fold for the C ring[12,17] and around 33-fold for the MS ring[16].

By cryoEM structural analyses of the flagellar basal body isolated from *Salmonella* cells and the MS ring formed by over-expressed full-length FliF of *Salmonella*, we determined the rotational symmetry of the native MS ring structure to be 34-fold with no variation (Fig. 1). This suggested that the symmetry variation in the recent high-resolution structures of the MS ring[16] was possibly produced by C-terminal truncations of FliF, as was indicated in the SDS-PAGE band pattern of their FliF preparation[16]. To examine this possibility, we prepared two C-terminally-truncated FliF fragments, $FliF_{1-503}$ and $FliF_{1-456}$ with 57 and 104 residues truncations, respectively, (Supplementary Fig. 8) and carried out cryoEM image analysis and 3D reconstruction of the MS rings formed by these fragments with C1 symmetry (Supplementary Figs. 9 and 10). The symmetry was 33-fold for the $FliF_{1-503}$ ring (EMDB ID: EMD-30940) and 34- and 35-fold for the $FliF_{1-456}$ ring (Supplementary Fig. 11) (EMDB ID: EMD-30942 and EMD-30941), which confirmed that the symmetry variation in the MS ring was caused by C-terminal truncations and that the full length FliF forms a ring of 34-fold symmetry.

We also carried out the symmetry analysis of the basal body C ring in a more direct manner than the previous study[12] by looking at enface views of the ring revealing subunit blobs (Supplementary Fig. 1c) and showed a symmetry variation to have a peak at 34-fold with much less populations of non 34-fold symmetry rings. So, the symmetries of the two rings are matched in the majority of the flagellar basal body albeit there are still minor populations having symmetry mismatches of one more or one less subunit in the C ring. The C ring assembly is thought to be initiated by co-folding of the C-terminal chain of FliF extending out on the circumference of the MS ring in the cytoplasm and the N-terminal domain of the FliG[31,32]. Also, mutation studies of *Salmonella* flagella have identified two types of FliF-FliG fusion proteins that are functional in motility[33]. These results predicted a rather strict 1:1 stoichiometry of FliF and FliG in the C ring assembly around the MS ring. However, our present structural analysis clearly shows symmetry mismatches between the two rings albeit rather small, suggesting a certain level of flexibility in FliG ring assembly around the MS ring to initiate the entire C ring assembly together with the other switch proteins FliM and FliN.

The structure of the S ring and cylindrical collar in the upper part of the MS ring is now resolved at high resolution (Fig. 2), and mutation analyses based on the atomic model show that conserved hydrophobic residues of FliF located at the intersubunit interface are responsible not only for MS ring formation by stabilizing intersubunit interactions but also for the assembly of the type III protein export apparatus and the rod within the MS ring (Fig. 3). Interestingly, the structures of the basal body and MS ring also showed 23-fold and 11-fold symmetries in the inner part of the M ring and a density feature at the center that looks exactly like the protein export gate complex formed by FliP, FliQ, and FliR[19] (Fig. 4). Contributing only 23 copies of RBM1-2 of total 34 to the formation of the inner core M ring would possibly be to form the central hole of an appropriate size for the protein export gate to efficiently assemble at the center of the MS ring. Thus, it is now clear that FliF is folded into two distinct conformations to build the MS ring structure with different symmetries in different parts for its multiple functions. This was also suggested by the recent high-resolution structures[16] albeit the symmetry variation and some of the structural features may not be physiologically relevant, possibly due to the C-terminal truncations of FliF as confirmed by our present analysis (Supplementary Fig. 11). The folding and assembly of 34 FliF subunits forming the multi-symmetry structure of the native MS ring will be described elsewhere. It is at least confirmed now that the MS ring and C ring are tightly bound to each other to act as a rotor unit of the flagellar motor to generate and transmit torque to the rod, hook and filament for bacterial motility.

## Methods

**Bacterial strains, plasmids and media**. Bacterial strains and plasmids used in this study are listed in Supplementary Table 1. L-broth contained 1% (w/v) Bacto-tryptone, 0.5% (w/v) Bacto-yeast extract and 0.5% (w/v) NaCl[28]. Soft tryptone agar plates contained 1% (w/v) Bacto-tryptone, 0.5% (w/v) NaCl and 0.35% Bacto agar (w/v)[34]. 2 × YT medium contained 1.6% (w/v) Bacto-tryptone, 1.0% (w/v) Bacto-yeast extract, 0.5% (w/v) NaCl.

**Purification of the basal body**. *Salmonella* HK1003 [*flgE*Δ(9-20) Δ*clpP*::Cm] (CCW motor) and TM022 [*flgE*Δ(9-20) Δ*clpP*::Cm *fliG*ΔPAA[35]] (CW motor) mutant cells, in which the number of the basal body was increased by the deletion of the ClpP protease[20], were grown in 5 l of L-broth at 37 °C until the cell had reached a late-logarithmic stage. The cells were harvested by centrifugation at 4600 × *g* for 10 min. The pellets were suspended in 240 ml of 50 mM Tris-HCl buffer at pH 8.0 containing 0.5 M sucrose in an ice bath. EDTA and lysozyme were added to final concentrations of 10 mM and 0.1 mg ml⁻¹, respectively, and the mixture was stirred for 30 min to convert cells to spheroplasts. Triton X-100 and MgSO₄ was added to the mixture to final concentrations of 1% and 10 mM, respectively. After stirring in the ice bath for 1 h, the mixture was centrifuged at 15,000 × *g* for 20 min to remove insoluble debris. An aliquot of 5 M NaOH was added to the supernatant to adjust pH to 10.5. The solution was centrifuged at 60,000 × *g* for 60 min, and the pellet was suspended in a buffer containing 10 mM Tris-HCl pH8.0, 5 mM EDTA and 1% Triton X-100. After repeating the above procedure twice, the basal body was purified by sucrose density gradient centrifugation at 68,000 × *g* for 14 h. Fractions with 20–50% sucrose were collected and centrifuged at 60,000 × *g* for 60 min. The pellet was suspended in Buffer S (25 mM Tris-HCl pH 8.0, 1 mM EDTA, 50 mM NaCl) containing 0.05% Triton X-100 and 0.05% LMNG or Buffer I (50 mM Tris-HCl pH 8.0, 50 mM NaCl, 25 mM imidazole, 0.05% Triton X-100 and 0.05% LMNG). At least ten times of independent purification were carried out.

**Overexpression of FliF and purification of the MS ring**. *E. coli* BL21 (DE3) strain carrying either pKOT112 (ref. [36]) (for dataset 1) or pKOT105 (ref. [37]) (for dataset 2) was grown overnight in 20 ml of a cell culture medium containing 0.1 g tryptone, 0.05 g yeast extract, 0.05 g NaCl, 50 μg ml⁻¹ of ampicillin and 30 μg ml⁻¹ of chloramphenicol in a shaker at 37 °C. A 20 ml of overnight culture was added to a 2 l of L-broth containing 50 μg ml⁻¹ of ampicillin and 30 μg ml⁻¹ of chloramphenicol and the cells were grown in a 37 °C orbital shaker (100 rpm) until the culture density reached an OD600 of 0.5–0.7. An aliquot of 0.5 M isopropyl β-D-1-thiogalactopyranoside (IPTG) was added to 2 L cell culture medium to a final concentration of 0.5 mM, and the growth was continued in a 30 °C orbital shaker (100 rpm) for 4 h. The cells were collected by centrifugation at 4600 × *g* for 10 min at 4 °C. Harvested cells were resuspended in 40 ml of a French press buffer (50 mM Tris-HCl pH 8.0, 5 mM EDTA-NaOH, 50 mM NaCl) containing a protease inhibitor cocktail (Complete, EDTA-free) and were disrupted using a French press at a pressure level of 10,000 psi. After cell debris and undisrupted cells were removed by centrifugation (20,000 × *g*, 20 min, 4 °C), the crude membrane fraction was isolated by ultracentrifugation (90,000 × *g*, 60 min, 4 °C). The pellet was solubilized in 40 ml of Alkaline buffer (50 mM CAPS-NaOH pH 11.0, 5 mM EDTA-NaOH, 50 mM

NaCl, 1% Triton X-100) by mechanical shearing with a 1 ml plastic syringe with 20 gauge needle and was incubated at 4 °C for 1 h. After insoluble material was removed by centrifugation (20,000 × *g*, 20 min, 4 °C), solubilized proteins were collected by ultracentrifugation (90,000 × *g*, 60 min, 4 °C). The pellet was resuspended in 3 ml of Buffer S containing 0.1% Triton X-100 by mechanical shearing as above, and the sample was loaded onto a 15–40% (w/w) continuous sucrose density gradient in Buffer C (10 mM Tris-HCl pH 8.0, 5 mM EDTA-NaOH, 1% Triton X-100) and spun by centrifugation in a swing rotor (49,100 × *g*, 13 h, 4 °C). Density fractions with a volume of 700 µl each were collected by a gradient fractionator (BIOCOMP, NB, Canada) and a fraction collector and analyzed by SDS-PAGE for a band of FliF. Peak fractions were collected, and the MS ring was further concentrated by ultracentrifugation (90,000 × *g*, 60 min, 4 °C). The pellet was resuspended in 30 µl of Buffer S containing 0.05% LMNG. At least ten times of independent purification were carried out.

**Sample vitrification and cryoEM data acquisition.** For the basal body sample, Quantifoil Mo 200 mesh R0.6/1.0 holey carbon grids (Quantifoil) were glow discharged on a glass slide for 10 s. A 2.5 µl aliquot of the sample solution was applied to the grid and blotted by a filter paper for 3 s × 2 times with 2 s drain time at 100% humidity and 4 °C. For the MS ring sample, Quantifoil Cu 200 mesh R0.6/1.0 holey carbon grids were glow discharged on a glass slide for 30 s. A 2.6 µl aliquot of the sample solution was applied onto the grid and blotted by a filter paper for 7 s at 100% humidity and 4 °C. The grids were quickly frozen by rapidly plunging into liquid ethane using a Vitrobot Mark III quick freezing device (Thermo Fisher Scientific). The grids were inserted into a Titan Krios transmission electron microscope (Thermo Fisher Scientific) operated at 300 kV, with the cryo specimen stage cooled with liquid nitrogen. CryoEM images were recorded with a Falcon II 4k × 4k CMOS direct electron detector (Thermo Fisher Scientific) at a nominal magnification of ×75,000 for the FliF ring data 1, corresponding to an image pixel size of 1.07 Å for high-resolution image analysis, or ×59,000 for the FliF ring data 2 and the basal body, corresponding to an image pixel size of 1.4 Å, using the EPU software package (Thermo Fisher Scientific). Movie frames were recorded at a dose rate of 45 e⁻/pix/sec over an exposure time of 2 s. The total accumulated dose of 90 e⁻/Å² was fractionated into seven frames. Image datasets of 1589 (dataset 1) and 806 micrographs (dataset 2) were collected for the FliF ring, and 1589 (for 2D classification) and 18,256 (for 3D reconstruction) micrographs for the basal body, using a defocus range between 1.0 and 3.0 µm (Supplementary Table 2).

**Symmetry analysis of the MS ring of the basal body.** Many of the basal body particles in cryoEM images showed end-on views. The particle images were picked up using an in-house particle picking program applying a deep learning method based on YOLO neural network[38]. About 2000 particles were manually picked from 200 micrographs and were used for a training of the neural network. In total 34,896 particles were extracted from 1578 micrographs. 2D classifications were carried out using Relion-2.1 (ref. [39]). After the second 2D classification, class average images showing the rivet-like structure with the MS ring and the rod were used for rotational symmetry analysis of the MS ring. The outer part of the ring images was converted from Cartesian to polar coordinates, the autocorrelation function was calculated, and the rotational symmetry was analyzed by Fourier transformation (Supplementary Fig. 1b).

**3D reconstruction of the basal body.** The basal body in Buffer I was used for 3D structural analysis of the basal body. The particle images were picked up using an in-house particle picking program as mentioned above. About 2000 particles were manually picked up from 200 micrographs and were used for training. In total 385,803 particles were extracted from 18,256 micrographs. 2D and 3D classifications and 3D reconstruction were carried out using Relion-2.1 (ref. [39]) or Relion-3.0 (ref. [40]). We used a 3D map of the basal body in a previous study[41] as the initial 3D model. After subtracting the C-ring density, 275,548 good particle images were used to construct 3D images in five classes. The two best 3D classes were individually refined and then merged for the final 3D refinement. The final 3D map was reconstructed using 149,341 particles at a resolution of 6.8 Å.

**Symmetry analysis of the C ring.** CryoEM images of the wild-type (CCW) and CW mutant basal bodies were collected using a JEM-3200FSC electron cryomicroscope (JEOL) equipped with a liquid-nitrogen cooled specimen stage, an Ω-type energy filter and a field-emission electron gun, operated at an accelerating voltage of 200 kV. The images were captured by an F415mp CCD camera (TVIPS) at a magnification of ×88,800 corresponding to a pixel size of 1.69 Å, a defocus range of 1.0–2.5 µm and an electron dose of 40 e⁻/Å⁻². Defocus and astigmatism of the images were determined using CTFFIND3 (ref. [42]). To estimate the symmetry of the C ring, end-on view images of the basal bodies were boxed out by BOXER[43]. The C ring part of each end-on view image was converted from the Cartesian to polar coordinates, the autocorrelation function was calculated, and the rotational symmetry was analyzed by Fourier transformation (Supplementary Fig. 1c). The end-on view images with distinct rotational symmetries were also classified, aligned, and averaged to show the C ring images with different symmetries in Supplementary Fig. 1c.

**Image processing of the MS ring formed by overexpressed FliF.** Image processing procedures of the MS ring are described in Supplementary Figs. 3 and 7. The movie frames were aligned to correct for beam-induced movement and drift by MotionCor2 (ref. [44]), and the parameters of the contrast transfer function (CTF) were estimated by Gctf[45].

For the FliF ring dataset 1 with the outer M ring part disordered, in total 339,861 particle images were automatically picked up from 1589 micrographs using Gautomatch (http://www.mrc-lmb.cam.ac.uk/kzhang/), and then 2D and 3D classifications were performed using Relion-2.1 (ref. [39]) and 3.0 (ref. [40]). Particle images from good 2D class average images were selected for the initial 3D model using CryoSPARC2 (ref. [46]). In total, 99,560 particles from the best 3D class were subjected to ab initio reconstruction and hetero refinement, and finally 38,889 particles from the best hetero refinement model were subjected to non-uniform refinement with C34 rotational symmetry using CryoSPARC2 (ref. [46]). The final refinement yielded a 3D map with a global resolution of 3.70 Å and a B factor of −197.5 Å² according to the 0.143 criterion of the Fourier shell correlation (FSC). The local resolution was estimated using CryoSPARC2 (ref. [46]). The processing strategy is described in Supplementary Fig. 3, and the model refinement statistics in Supplementary Table 2.

For the FliF ring dataset 2 with the M ring part well ordered, in total 156,459 particle images were automatically picked from 806 micrographs using Gautomatch (http://www.mrc-lmb.cam.ac.uk/kzhang/), and 2D and 3D classifications were performed using Relion-2.1 (ref. [39]). Particle images from good 2D class average images were selected for the initial 3D model using CryoSPARC2 (ref. [46]). In total, 18,883 particles from the best 3D class were subjected to 3D refinement with C1, C11 and C34 rotational symmetry using Relion-2.1 (ref. [39]). The final refinement and postprocessing yielded a 3D map with a global resolution and a B-factor of 12 Å and −35 Å² for C1, 9.0 Å and −495 Å² for C11 and 7.4 Å and −450 Å² for C34, according to the 0.143 criterion of the FSC. The processing strategy is described in Supplementary Fig. 7, and the model refinement statistics in Supplementary Table 2.

**Model building and refinement of the S ring structure.** The atomic model of the S ring was constructed using Coot[47] and refined with real space refinement based on the cryoEM map (EMD-30612) using Phenix[48] under NCS constrains of 34-fold rotational symmetry and secondary structure restraints. The refinement statistics are summarized in Supplementary Table 2.

**Site-directed mutagenesis.** Site-directed mutagenesis was carried out using the QuickChange site-directed mutagenesis method (Stratagene). All of the *fliF* mutations were confirmed by DNA sequencing. DNA sequencing reactions were carried out using BigDye v3.1 (Applied Biosystems) and then the reaction mixtures were analyzed by a 3130 Genetic Analyzer (Applied Biosystems). Plasmids and all primers used in this study are listed in Supplementary Table 1 and 4.

**Motility assays.** We transformed a *Salmonella fliF* null mutant strain, TH12415 (Δ*fliF*), with pET3c-based plasmids, and fresh transformants were inoculated onto soft tryptone agar plates containing 100 µg ml⁻¹ ampicillin and incubated at 30 °C.

**Fractionation of cell membrane.** *Salmonella* TH12415 cells harboring an appropriate plasmid were grown exponentially in 30 ml L-broth containing 100 µg ml⁻¹ ampicillin at 30 °C with shaking. The cells were harvested, resuspended in 3 ml PBS, and sonicated. After the cell debris was removed by low-speed centrifugation, the cell lysates were ultracentrifuged (100,000 × *g*, 60 min, 4 °C). After carefully removing the soluble fractions, membranes were resuspended in 300 µl of sodium dodecyl sulfate (SDS) loading buffer [62.5 mM Tris-HCl, pH 6.8, 2% (w/v) SDS, 10% (w/v) glycerol, 0.001% (w/v) bromophenol blue] containing 1 µl of 2-mercaptoethanol and heated at 95 °C for 3 min. Proteins were separated by SDS–polyacrylamide gel electrophoresis (SDS-PAGE) and transferred to nitrocellulose membranes (Cytiva) using a transblotting apparatus (Hoefer). Then, immunoblotting with polyclonal anti-FliF or anti-FliN antibody was carried out. The primary antibody was diluted 10,000-fold and the secondary antibody was diluted 5000-fold using TBS-T buffer (20 mM Tris-HCl, pH 7.5, 500 mM NaCl, 0.1%(v/v) Tween-20). An ECL prime immunoblotting detection kit (GE Healthcare) was used to detect target bands. Chemiluminescence signals were detected by a Luminoimage analyzer, LAS-3000 (GE Healthcare). Three independent experiments were performed.

**Flagellar protein export assay.** *Salmonella* cells were grown in L-broth containing 100 µg ml⁻¹ ampicillin at 30 °C with shaking until the cell density had reached an OD₆₀₀ of ca. 1.4–1.6. Cultures were centrifuged to obtain cell pellets and culture supernatants. The cell pellets were resuspended in SDS loading buffer solution containing 1 µl of 2-mercaptoethanol. Proteins in the culture supernatants were precipitated by 10% trichloroacetic acid and suspended in a Tris/SDS loading buffer (one volume of 1 M Tris, nine volumes of 1 × sample buffer solution)[49] containing 1 µl of 2-mercaptoethanol. Both whole cellular proteins and culture supernatants were normalized to a cell density of each culture to give a constant number of *Salmonella* cells. After SDS-PAGE, immunoblotting with polyclonal anti-FlgD, anti-FlgE or anti-FliC antibody was performed. At least three independent assays were carried out.

**In vivo disulfide crosslinking**. *Salmonella* TH12415 cells harboring pMKMiF001, pMKMiF008, pMKMiF009 or pMKMiF011 were exponentially grown in L-broth containing 100 μg ml$^{-1}$ ampicillin at 30 °C with shaking. Aliquots of the cultures containing a constant number of cells were centrifuged to harvest the cells. Membrane fractions were prepared from the above *Salmonella* cells, and disulfide crosslinking were induced by adding 1 μl of 20 mM iodine in 99% ethanol[50,51]. The membrane fractions were gently shaken for 5 min at room temperature, and then oxidation was quenched by adding 2 μl of 1 M N-ethylmaleimide in 99% ethanol. After gentle shaking for 5 min at room temperature, the samples were mixed with SDS loading buffer and heated at 95 °C for 3 min. After SDS-PAGE using 4–15% Mini-PROTEAN TGX Precast gels (Bio-Rad), immunoblotting with polyclonal anti-FliF antibody was carried out. The primary antibody was diluted 10,000-fold and the secondary antibody was diluted 5000-fold using TBS-T buffer. Six independent experiments were performed.

**Purification of C-terminally-truncated fragments of FliF, FliF$_{1-503}$ and FliF$_{1-456}$.** A 15 ml solution of the overnight culture of *Salmonella* SJW1368 [Δ(*cheW-flhD*)] cells transformed with pMKMiF13 (pTrc99AFF4/FliF$_{1-503}$) or pMKMiF14 (pTrc99AFF4/FliF$_{1-456}$) were inoculated into a 1.5 l of fresh 2 × YT medium containing 100 μg ml$^{-1}$ ampicillin and were grown at 30 °C until the cell density had reached an OD$_{600}$ of about 0.6. After 30 min incubation at 4 °C, the cells were grown at 16 °C for 12 h. SJW1368 cells overexpressing either FliF$_{1-503}$ or FliF$_{1-456}$ were harvested by centrifugation (6400 g, 10 min, 4 °C) and stored at −80 °C. The cells were thawed, resuspended in 55 ml of French press buffer and disrupted by passage through a FRENCH pressure cell (FA-032, Central Scientific Commerce). The cell lysates were centrifuged (20,000 g, 15 min, 4 °C) to remove undisrupted cells. The supernatants were ultracentrifuged (90,000 g, 1 h, 4 °C). The harvested membranes were suspended in 40 ml of Alkaline buffer and incubated at 4 °C for 1 h. After centrifugation (20,000 g, 20 min, 4 °C), the supernatants were ultracentrifuged (90,000 g, 60 min, 4 °C), and the pellets were resuspended in 10 mM Tris-HCl, pH 8.0, 5 mM EDTA, 1% Triton-X100 and incubated at 4 °C for 1 h. The solution was loaded to a 15–40% (w/w) sucrose density gradient in 10 mM Tris-HCl, pH 8.0, 5 mM EDTA, 1% Triton-X100. After ultracentrifugation (49,100 g, 13 h, 4 °C), fractions containing FliF$_{1-503}$ or FliF$_{1-456}$ were collected (Supplementary Fig. 8) and diluted by seven times volume of Buffer S containing 0.1% Triton-X100 and ultracentrifuged (90,000 g, 60 min, 4 °C). The pellets were then resuspended in 30 μl of Buffer S containing 0.1% Triton-X100.

**CryoEM data acquisition and image processing of the FliF rings formed FliF$_{1-503}$ and FliF$_{1-456}$.** For both FliF$_{1-503}$ and FliF$_{1-456}$ datasets, Quantifoil Cu 200 mesh R0.6/1.0 holey carbon grids (Quantifoil) were glow discharged for 20 s. A 2.7 μl aliquot of the sample solution was applied to the grid and blotted by a filter paper for 3 s at 100% humidity and 4 °C. The grids were quickly frozen by rapidly plunging into liquid ethane using a Vitrobot Mark III quick freezing device (Thermo Fisher Scientific). The grids were inserted into a CRYO ARM 300 transmission electron cryomicroscope (JEOL) operated at 300 kV, with the cryo specimen stage cooled with liquid nitrogen. CryoEM images were recorded with a K3 direct electron detector (Gatan) at a nominal magnification of ×50,000, corresponding to an image pixel size of 1.0 Å, using SerialEM[52]. Movie frames were recorded at a dose rate of 50 e$^{-}$ pix$^{-1}$ sec$^{-1}$ with an exposure time of 2.5 s. Datasets of 6566 and 6200 images were collected for FliF$_{1-503}$ and FliF$_{1-456}$, respectively, using a defocus range between 1.0 and 2.5 μm (Supplementary Table 3).

For both the FliF$_{1-503}$ and FliF$_{1-456}$ datasets, correction for beam-induced movement and drift, the estimation of the CTF and particles picking were performed by Warp (ref. [53]). In total, 226,068 and 446,123 particle images were automatically picked, respectively. Particle images from good 2D class average images were selected for the initial 3D reconstruction using Relion-3.1 (ref. [40]). In total, 99,858 and 262,434 particles, respectively, were subjected to 3D classification with C1 symmetry into three classes using Relion-3.1 (ref. [40]) as shown in Supplementary Figs. 9 and 10. For FliF$_{1-456}$, one of the two good 3D classes had C34 symmetry and another one had C35 symmetry. After 3D refinement for each of these two classes, postprocessing yielded 3D maps with resolutions of 4.46 Å and 4.59 Å according to the 0.143 criterion of the FSC, respectively. For FliF$_{1-503}$, a good 3D class having C33 symmetry proceeded 3D refinement and postprocessing yielding for a 3D map with a resolution 6.90 Å according to 0.143 criterion of the FSC. All process was carried out with C1 symmetry (Supplementary Table 3). The image processing strategy is described in Supplementary Figs. 9 and 10, respectively.

**Reporting summary**. Further information on research design is available in the Nature Research Reporting Summary linked to this article.

## Data availability
The cryoEM volumes have been deposited in the Electron Microscopy Data Bank under accession code EMD-30612 for the MS ring (dataset 1), EMD-30613 for the basal body, EMD-30360, EMD-30361 and EMD-30363 for the MS ring (dataset 2) with C1, C11 and C34 symmetry imposed, respectively, EMD-30940 for the 33-fold symmetry MS ring formed by FliF$_{1-503}$, EMD-30941 and EMD-30942 for the 35- and 34-fold symmetry MS rings formed by FliF$_{1-456}$, respectively. The atomic coordinates for the S ring and collar of

the MS ring have been deposited in the Protein Data Bank under accession code 7D84 [https://doi.org/10.2210/pdb7D84/pdb]. Other data are available from the corresponding author upon reasonable request. Source data are provided with this paper.

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

## Acknowledgements

We thank Kelly T. Hughes for his kind gift of a *Salmonella fliF* null mutant, Tomoko Yamamoto for a phage for *clpP* deletion, Hideyuki Matsunami for *Salmonella* HK1003 mutant strain and Noriyuki Takekawa for help preparing a figure. This research was supported in part by JSPS KAKENHI Grant Numbers 25000013 to K.N., 18K06155 and 20770083 to T.Miyata, 18K14639 to A.K., JP26293097 and JP19H03182 to T.M.inamino, JP15H05593 and JP20K15749 to M.K., JP15H02386 to K.I., and MEXT KAKENHI Grant Numbers JP15H01640 and JP20H05532 to T.Minamino. This research was also supported by Platform Project for Supporting Drug Discovery and Life Science Research (BINDS) from AMED under Grant Number JP19am0101117 to K.N., by the Cyclic Innovation for Clinical Empowerment (CiCLE) Grant Number JP17pc0101020 from AMED to K.N. and by JEOL YOKOGUSHI Research Alliance Laboratories of Osaka University to K.N.

## Author contributions

T.K. and K.N. conceived the project; A.K., T.K., T.Minamino and K.N. designed experiments; A.K., T.Miyata and M.K. prepared the samples for cryoEM; T.K. set up cryoEM imaging system including both hardware and software; A.K., T. Miyata, F.M. and T.K. collected and analyzed cryoEM image data; M.K. and T.Minamino performed genetic, biochemical and physiological experiments; A.K. and K.I. built the atomic model; All authors studied the atomic model wrote the paper based on discussion with other authors.

## Competing interests

The authors declare no competing interests.
