## [Peer Review File · Nature Communications]

REVIEWER COMMENTS

Reviewer #1 (Remarks to the Author):

This paper describes the structure of the bacterial flagellar motor protein FliF. FliF forms the MS-ring, the hub of the flagellar motor, which surrounds the export apparatus and rod, templates formation of the C-ring, and mediates interface of various symmetry mismatches within the motor. The most important advance of this work is purification of FliF from assembled flagellar motors instead of from overexpression alone, as published recently by Susan Lea and colleagues, who saw varied symmetries from 32-fold to 36-fold. The author's purification protocol provides MS-rings exclusively with 34-fold symmetry in the more peripheral structures, indicating that 34 FliF protomers comprise the native MS-ring. Surprisingly, however, the authors go on to show that even MS-rings purified from overexpressed FliF, in their hands, have exclusively 34-fold symmetry. The new finding of the paper is that the MS-ring is 34-fold symmetric, and that the discrepancy with Lea's work may be due to Lea's truncation of the C-terminus of FliF.

The paper is reasonably well-written and tackles the central protein of the bacterial flagellar motor. The authors have carried out substantial work and collected a wealth of data, and their conclusions are well-argued.

Nevertheless I have a few major concerns:

- I didn't find explanations for the discrepancies with Lea convincing. This is a major concern, as this is the major new contribution of this new study. Lea used similar numbers of particles (a few more) to discover a diversity of symmetries. The author's claim that these discrepancies are "possibly produced by the C-terminal truncations of FliF" is plausible, but given that the paper's novelty rests upon this discrepancy, I consider testing this speculation to be crucial to rule out the null hypothesis that Lea's data is simply better able to discriminate other symmetries that the authors are not capable of. Confirming they can reproduce Lea's results by truncation of the C-terminus is therefore critical for me to have confidence in the work.
- Indeed, the study feels incomplete, particularly in light of the sentence in the Discussion: "The folding and assembly of 34 FliF subunits forming the multi-symmetry structure of the native MS ring will be described elsewhere.". The analysis and discussion are shallow relative to Lea's recent work. See "Minor points" for a few suggestions.

MINOR POINTS

- Grammar needs checking throughout, including even the very first sentence, "Bacteria actively swim in liquid environments by rotating long, helical filamentous organelle called the flagellum.", and:
 - line 68: "For a long time until recently..."
 - line 73: "This puzzle was sort of resolved by the..."
 - line 79: "Also, the MS ring structures were analyzed for those formed by overexpressed FliF with some C-terminal truncations". ("some C-terminal truncations" seems really vague and kind of flippant to me)
 - line 120: "So the MS ring structure in the native flagellar basal body is composed of 34 FliF subunits."
 - line 304: "This puzzle was sort of resolved by the recent high-resolution structure..."
- Class averages in Figure 1 are low quality and saturated.
- I would be very interested in hearing the author's thoughts on the implication of the apparent 23-fold symmetry mismatch with ~6-fold symmetry of FliPQR, Previous work from Jun Liu convincingly demonstrated that injectisome FliF homolog/analog SctDJ is 24-fold symmetric to match the ~6-fold symmetry of SctRSTU -- but only in the presence of SctRSTU. I would have expected 24-fold symmetry of the FliPQR binding region based on this result. Do the authors have a view on why this is not the case?
- I would also be interested in hearing the author's view on the implications of their structure for the location of the estimated 9-fold symmetry of FliE? (Müller, et al . 'Characterization of the FliE Genes of

Escherichia Coli and Salmonella Typhimurium and Identification of the FliE Protein as a Component of the Flagellar Hook-Basal Body Complex.' Journal of Bacteriology 174, no. 7 (1992): 2298–2304. <https://doi.org/10.1128/jb.174.7.2298-2304.1992.>)

Reviewer #2 (Remarks to the Author):

The authors in this manuscript used cryoEM as a major tool to determine native structure of the bacterial flagellar MS ring, which is the core of the flagellar motor and provides base for the assembly of other major flagellar components. The study provided structural evidence that the MS ring shares similar symmetry as the flagellar C-ring, which is responsible for flagellar rotation switch. Importantly, the authors showed that the MS ring adopts two distinct conformations with 23-fold and 11-fold sub-symmetries. The distinct conformations enable the MS-ring to function as the template for the assembly of the C-ring, rod, and export apparatus. Furthermore, mutational analyses were used to map important residues that are responsible for inter-subunit interactions. Overall this is a comprehensive study on the core of the flagella, providing a better understanding of the assembly of the complex machine. However, this manuscript did not provide adequate discussion and detailed comparison between this study and the recent MS-ring paper (reference 16).

Specific comments:

1. In the abstract and text “the native MS ring structure formed by full-length FliF is 34-fold with no symmetry variation whereas the C ring has a small symmetry variation”. Symmetry analyses appear to be challenging for many cryoEM structures. 34-fold symmetry appears to be solid based on the class averages. However, given that the symmetric features are less obvious in Fig. 1c, it may not be possible to rule out other possible symmetries with less particles. Therefore, it will require additional evidence to support “no symmetry variation”.
2. What’s the difference between the cryoEM structure of the MS ring formed by overexpressed FliF and the previous cryoEM structure of the MS ring (reference 16).
3. It is not clear how the 3D reconstruction of the basal body at about 6.8 Å resolution (line 270) was generated. Both 23-fold and 11-fold sub-symmetries (Fig. 4) are not as obvious as 34-fold symmetry in Fig. 2. What is the resolution estimated for these maps?
4. In page 10, “The recent high-resolution structures of the MS ring with 33-fold symmetry with some variation would therefore be an artifact, possibly produced by the C-terminal truncations of FliF16”. Without additional evidence, it might not be a good idea to suggest that some variation in symmetry is an artifact.

Reviewer #3 (Remarks to the Author):

This paper describes a structural study of the flagellar MS-ring that is the core structure of the basal body, and also the C-ring that contains proteins that function in the generation of torque and in direction switching. The study is thorough, and casts important light on the symmetries of these structures, specifically showing that there is some symmetry mismatch, with the MS-ring adhering strictly to 34-fold symmetry and the C-ring displaying some variation in symmetry. The structure must therefore be capable of tolerating some variability in the number of C-ring subunits relative to MS-ring subunits. Such symmetry mismatch has been discussed in the literature for some time, and the present results provide important new information on this mismatch. I have a few minor comments and suggestions.

1. The results show that the structure has the ability to tolerate small symmetry variation. During the process of C-ring remodeling, which is suggested to take place to enable motors to switch direction under varying conditions (for example, variations in the level of the CW-signaling protein CheY), the number of FliM subunits is believed to vary a great deal. This would also imply some ability to tolerate

symmetry mismatch, in fact a much greater mismatch than demonstrated here. This topic (remodeling) merits some discussion in the paper. It also suggests that additional experiments to examine the C-ring symmetry in cells under conditions that should favor large or small C-rings (which is thought to depend on the CW bias) would be informative.

2. line 61. I wasn't sure why the protein FliE is not also mentioned here, as it is also a component of the rod.

3. line 74 "sort of" is used to characterize the degree to which a previous structural study of the MS-ring settled questions of symmetry mismatch. I felt that this usage may be too casual—the previous study arrived at results that were substantially similar to those here, yet with certain, significant, differences. As well as being rather casual-sounding, "sort of" might be selling the previous study a bit short.

4. line 86. I think the characterization of the PQR location as 'likely' understates what is known. I think it's pretty clear. Also, the helical nature of the PQR assembly is well established.

5. line 102. The word 'now' I gather indicates what is shown in the present work, but I wasn't sure of this at first. It could be made clearer by use of something more typical like "In the present work, it is shown that..'

6. line 138 The data indicate that the populations of non-34 symmetries is just somewhat higher, not actually much higher.

7. line 189. A structural difference from the ring of the T3SS ring is noted, involving the D2 domain of the protein PrgH, but from the information provided, it's not clear what this difference is or how it might be significant.

8. line 206. where the hydrophobic residues are introduced, it might be worth noting explicitly that these side-chains, though hydrophobic, are surface-exposed on the individual subunits (which I gather from the statement that they form the interfaces). How much hydrophobic contact area do they account for at the interface between subunits?

9. Line 227. Do these two mutants produce the same number of hook-basal bodies as the wild type, or is the number of HBBs diminished? If so, please note. If not, then it remains possible that they disfavor assembly of the MS ring—not enough to prevent assembly altogether, but enough to decrease the number of HBBs, and therefore, export apparatuses.

10. line 240. Is it known whether localization of FliN requires the formation of normal C-rings? Is it possible that the mere presence of FliF, FliG, and FliM, whether they assemble into rings or not, can enable association of some FliN with the membrane?

11. line 249. I was surprised to read here that two of the mutated positions are actually not at the interface between subunits. In the remarks introducing this mutational analysis section (lines 205-207), it seemed to imply that the aim of the mutations was to probe hydrophobic residues at the inter-subunit interface.

11. If we grant that HBBs are made in normal numbers in the mutants, and that all of them contain normal C-rings, then it seems that assembly of the export apparatus into the mutant assemblies must be normal, because construction of the rod and hook depends on the action of the export apparatus. It might be argued that what's impaired is the export of the later subunits that form the filament; i.e., the FliF mutations are somehow selectively impeding export of the 'late' cargoes.

12. line 261. It is suggested that the crosslinked MS ring can no longer have the type III secretion apparatus inserted. Was this (or could this be) tested by export assays like those described above for the other mutants?

13. line 306. The emphasis is on symmetry, which is certainly important, but I also wondered—given that the authors are saying that the PQR assembly with 5.5 subunits per turn is inserted into the central part of the MS ring that has 23-fold symmetry—whether the inner surface of the structure, which must mate to the PQR assembly, is smooth or rough. Does the 23-fold symmetry give rise to rough topography with 23-fold symmetry, or is it a fairly smooth structure, in which the symmetry is not really very important?

14. line 314. Same point as above regarding 'sort of'.

15. line 353. This seems an unnecessarily harsh critique of the earlier study. Is there really any basis to say that all detailed features of the previous study were non-physiological?

16. Fig. 2.a. This looks like a single averaged map, not a set of 2D class averages.

Response to the reviewers' comments:

To Reviewer #1:

This paper describes the structure of the bacterial flagellar motor protein FliF. FliF forms the MS-ring, the hub of the flagellar motor, which surrounds the export apparatus and rod, templates formation of the C-ring, and mediates interface of various symmetry mismatches within the motor. The most important advance of this work is purification of FliF from assembled flagellar motors instead of from overexpression alone, as published recently by Susan Lea and colleagues, who saw varied symmetries from 32-fold to 36-fold. The author's purification protocol provides MS-rings exclusively with 34-fold symmetry in the more peripheral structures, indicating that 34 FliF protomers comprise the native MS-ring. Surprisingly, however, the authors go on to show that even MS-rings purified from overexpressed FliF, in their hands, have exclusively 34-fold symmetry. The new finding of the paper is that the MS-ring is 34-fold symmetric, and that the discrepancy with Lea's work may be due to Lea's truncation of the C-terminus of FliF.

The paper is reasonably well-written and tackles the central protein of the bacterial flagellar motor. The authors have carried out substantial work and collected a wealth of data, and their conclusions are well-argued.

Re: Thank you for the comment recognizing the hard work on the important subject.

Nevertheless I have a few major concerns:

- I didn't find explanations for the discrepancies with Lea convincing. This is a major concern, as this is the major new contribution of this new study. Lea used similar numbers of particles (a few more) to discover a diversity of symmetries. The author's claim that these discrepancies are "possibly produced by the C-terminal truncations of FliF" is plausible, but given that the paper's novelty rests upon this discrepancy, I consider testing this speculation to be crucial to rule out the null hypothesis that Lea's data is simply better able to discriminate other symmetries that the authors are not capable of. Confirming they can reproduce Lea's results by truncation of the C-terminus is therefore critical for me to have confidence in the work.

Re: Thank you for the very reasonable comment. According to your suggestion, we prepared two FliF fragments, FliF₁₋₅₀₃ and FliF₁₋₄₅₆, by overexpressing C-terminally-truncated FliF with truncation of 57 and 104 residues, respectively (Supplementary Fig. 8), and carried out cryoEM image analysis of the MS ring formed by these FliF fragments with C1 symmetry (Supplementary

Figs. 9 and 10). The structures of these MS rings showed that the symmetry was 33-fold for the FliF₁₋₅₀₃ ring (EMDB ID: EMD-30940) and 34- and 35-fold for the FliF₁₋₄₅₆ ring (Supplementary Fig. 11), which confirmed that the symmetry variation in the MS ring was caused by C-terminal truncations and that the full length FliF forms a ring of 34-fold symmetry. We described this in the second paragraph of page 10.

- Indeed, the study feels incomplete, particularly in light of the sentence in the Discussion: "The folding and assembly of 34 FliF subunits forming the multi-symmetry structure of the native MS ring will be described elsewhere.". The analysis and discussion are shallow relative to Lea's recent work. See "Minor points" for a few suggestions.

Re: The main point of this paper is to discuss the symmetry and subsymmetries of the flagellar MS ring in the native form. The data we added in the revised manuscript as described above confirmed our point and made the study complete. I hope you agree.

Minor point

- Grammar needs checking throughout, including even the very first sentence, "Bacteria actively swim in liquid environments by rotating long, helical filamentous organelle called the flagellum."

Re: We inserted "a" in the sentence as "... by rotating a long, helical filamentous organelle...".

- line 68: "For a long time until recently..."

Re: We are not sure what is wrong with this phrase.

- line 73: "This puzzle was sort of resolved by the..."

Re: We rephrased this to "This puzzle seemed resolved..."

- line 79: "Also, the MS ring structures were analyzed for those formed by overexpressed FliF with some C-terminal truncations". ("some C-terminal truncations" seems really vague and kind of flippant to me)

Re: We rephrased this sentence as "Also, the MS ring structures were analyzed with those formed by overexpressed FliF¹⁶, possibly with some C-terminal truncations as suggested in the SDS-PAGE band pattern of the FliF protein preparation, and this may have produced the observed symmetry variation." to make it more specific and concrete.

- line 120: "So the MS ring structure in the native flagellar basal body is composed of 34 FliF subunits."

Re: We are not sure what is the problem with this sentence.

- line 304: "This puzzle was sort of resolved by the recent high-resolution structure..."

Re: We rephrased this as "This puzzle appeared to have been resolved by the recent high-resolution structure..."

- Class averages in Figure 1 are low quality and saturated.

Re: We apologize for the low-quality presentation of the class averages. We adjusted the grey scale to visualize the symmetry much clearer and put number labels around the ring for easy counting of 34 subunits.

- I would be very interested in hearing the author's thoughts on the implication of the apparent 23-fold symmetry mismatch with ~6-fold symmetry of FliPQR, Previous work from Jun Liu convincingly demonstrated that injectisome FliF homolog/analog SctDJ is 24-fold symmetric to

match the ~6-fold symmetry of SctRSTU -- but only in the presence of SctRSTU. I would have expected 24-fold symmetry of the FliPQR binding region based on this result. Do the authors have a view on why this is not the case?

Re: We do not think that the 24-fold symmetry of the injectisome SctDJ ring matches the ~6-fold symmetry of the SctRSTU gate complex, partly because the SctRSTU gate complex is not a ring but a helical assembly. We think that the important structural feature of the inner part of the M ring formed by 23 subunits of RBM1-2 of FliF is not its 23-fold symmetry but to form the central hole of a size appropriate for gate complex assembly inside it. We stated this in the last paragraph of Discussion as "Contributing only 23 copies of RBM1-2 of total 34 to the formation of the inner core M ring would possibly be to form the central hole of an appropriate size for the protein export gate to efficiently assemble at the center of the MS ring."

- I would also be interested in hearing the author's view on the implications of their structure for the location of the estimated 9-fold symmetry of FliE? (Müller, et al. 'Characterization of the FliE Genes of Escherichia Coli and Salmonella Typhimurium and Identification of the FliE Protein as a Component of the Flagellar Hook-Basal Body Complex.' Journal of Bacteriology 174, no. 7 (1992): 2298–2304. <https://doi.org/10.1128/jb.174.7.2298-2304.1992>).

Re: In the cryoEM structure of the basal body, we identified five FliE subunits forming a helical assembly directly above the FliPQR gate complex inside the MS ring. We are writing up a manuscript describing the basal body structure including this. Susan Lea's group also identified such structure of FliE and described it in a preprint submitted to bioRxiv (<https://doi.org/10.1101/2020.12.05.413195>).

To Reviewer #2:

The authors in this manuscript used cryoEM as a major tool to determine native structure of the bacterial flagellar MS ring, which is the core of the flagellar motor and provides base for the assembly of other major flagellar components. The study provided structural evidence that the MS ring shares similar symmetry as the flagellar C-ring, which is responsible for flagellar rotation switch. Importantly, the authors showed that the MS ring adopts two distinct conformations with 23-fold and 11-fold sub-symmetries. The distinct conformations enable the MS-ring to function as the template for the assembly of the C-ring, rod, and export apparatus. Furthermore, mutational analyses were used to map important residues that are responsible for inter-subunit interactions. Overall this is a comprehensive study on the core of the flagella, providing a better understanding of the assembly of the complex machine.

Re: Thank you for your fair and kind judgement of our study and helpful comments for revising the manuscript.

However, this manuscript did not provide adequate discussion and detailed comparison between this study and the recent MS-ring paper (reference 16).

Re: Thank you also for your critical comment. We admit that we did not describe detailed comparison of the MS ring structures between ours and those presented in the recent paper (ref. 16). This is partly because the atomic models of the S ring and collar parts of the MS ring were nearly identical between the two, as we stated in the main text as "The model was nearly identical to the corresponding part of the recent MS ring structure with 34-fold symmetry¹⁶". In this paper, we rather wanted to focus on the symmetry and subsymmetries of the flagellar MS ring in the native form.

We therefore prepared two C-terminally-truncated FliF, carried out cryoEM image analysis of the MS rings formed by these two FliF fragments and confirmed that the symmetry variation of the MS ring presented in the recent paper (ref. 16) was produced by unfortunate C-terminal truncations of FliF in their preparation. We presented these data in Supplementary Figs. 8 – 11 and described the results in Discussion.

Specific comments:

1. In the abstract and text “the native MS ring structure formed by full-length FliF is 34-fold with no symmetry variation whereas the C ring has a small symmetry variation”. Symmetry analyses appear to be challenging for many cryoEM structures. 34-fold symmetry appears to be solid based on the class averages. However, given that the symmetric features are less obvious in Fig. 1c, it may not be possible to rule out other possible symmetries with less particles. Therefore, it will require additional evidence to support “no symmetry variation”

Re: We apologize for the low contrast images of the class averages. We adjusted the grey scale to visualize the symmetry much clearer and put number labels around the ring for easy counting of 34 subunits. We hope this is satisfactory.

2. What's the difference between the cryoEM structure of the MS ring formed by overexpressed FliF and the previous cryoEM structure of the MS ring (reference 16).

Re: Our structure of the MS ring formed by overexpressed full-length FliF is nearly identical to the 34-fold symmetry MS ring structure presented in ref. 16, as we stated above and in the main text.

3. It is not clear how the 3D reconstruction of the basal body at about 6.8 Å resolution (line 270) was generated. Both 23-fold and 11-fold sub-symmetries (Fig. 4) are not as obvious as 34-fold symmetry in Fig. 2. What is the resolution estimated for these maps?

Re: Please refer to "3D reconstruction of the basal body" in the Method section and Supplementary Fig. 6 as to how the 3D map was generated. The resolution of the entire basal body map is 6.8 Å, which was determined by Fourier shell correlation at 0.143 as shown in Supplementary Fig. 6. We think that the 23 subunits can be easily counted in the slice images of the inner M ring (labeled “core”) presented in Fig. 4b and that the 11-fold symmetry is clearly visible in the 2D class average image of the MS ring shown in Fig. 4d.

4. In page 10, “The recent high-resolution structures of the MS ring with 33-fold symmetry with some variation would therefore be an artifact, possibly produced by the C-terminal truncations of FliF16”. Without additional evidence, it might not be a good idea to suggest that some variation in symmetry is an artifact.

Re: We now provided additional evidence in Supplementary Figs. 8 – 11 to confirm it. We hope this is satisfactory.

To Reviewer #3:

This paper describes a structural study of the flagellar MS-ring that is the core structure of the basal body, and also the C-ring that contains proteins that function in the generation of torque and in direction switching. The study is thorough, and casts important light on the symmetries of these structures, specifically showing that there is some symmetry mismatch, with the MS-ring adhering strictly to 34-fold symmetry and the C-ring displaying some variation in symmetry. The structure must therefore be capable of tolerating some variability in the number of C-ring subunits relative to MS-ring subunits. Such symmetry mismatch has been discussed in the literature for some time, and the present results provide important new information on this mismatch. I have a few minor comments and suggestions.

Re: Thank you for your recognition on the implications of our study and many helpful comments for improving the manuscript.

1. The results show that the structure has the ability to tolerate small symmetry variation. During the process of C-ring remodeling, which is suggested to take place to enable motors to switch direction under varying conditions (for example, variations in the level of the CW-signaling protein CheY), the number of FliM subunits is believed to vary a great deal. This would also imply some ability to tolerate symmetry mismatch, in fact a much greater mismatch than

demonstrated here. This topic (remodeling) merits some discussion in the paper. It also suggests that additional experiments to examine the C-ring symmetry in cells under conditions that should favor large or small C-rings (which is thought to depend on the CW bias) would be informative.

Re: Thank you for your suggestion. However, it is rather difficult to make a meaningful discussion on the large difference in the number of C ring subunits between the structure we see by cryoEM analysis of isolated basal body and those measured in cells by fluorescence microscopy. The C ring appears to be a stably closed ring, keeping the same number of subunits 34 even upon switching of motor rotation between CW and CCW. Besides, if the number of subunits increases to over 40 as observed by fluorescence microscopy, the distance between the MS ring and C-ring becomes too large for the C-terminal of FliF and the N-terminal of FliG to be connected and co-folded. Therefore, it is difficult to imagine that the C ring actually becomes significantly larger by the increase in the subunit number. But this kind of speculations does not seem to fit into this paper.

2. line 61. I wasn't sure why the protein FliE is not also mentioned here, as it is also a component of the rod.

Re: We included FliE in the description but as an adaptor protein for rod assembly.

3. line 74 "sort of" is used to characterize the degree to which a previous structural study of the MS-ring settled questions of symmetry mismatch. I felt that this usage may be too casual—the previous study arrived at results that were substantially similar to those here, yet with certain, significant, differences. As well as being rather casual-sounding, "sort of" might be selling the previous study a bit short.

Re: Thank you for the advice. We rephrased it as "This puzzle seemed resolved...".

4. line 86. I think the characterization of the PQR location as 'likely' understates what is known. I think it's pretty clear. Also, the helical nature of the PQR assembly is well established.

Re: Since the structure of the basal body is not yet available in literature, there is no evidence for the exact location of the PQR complex. We rephrased it as "thought to be".

5. line 102. The word 'now' I gather indicates what is shown in the present work, but I wasn't sure of this at first. It could be made clearer by use of something more typical like "In the present work, it is shown that.."

Re: We deleted "now" from the sentence.

6. line 138 The data indicate that the populations of non-34 symmetries is just somewhat higher, not actually much higher.

Re: The population of non 34-fold symmetry C rings was 56% in the previous study and 35% (CCW) and 34% (CW) in the present study. We do not think 56% is just somewhat higher than 35% or 34%.

7. line 189. A structural difference from the ring of the T3SS ring is noted, involving the D2 domain of the protein PrgH, but from the information provided, it's not clear what this difference is or how it might be significant.

Re: To make the difference clear, we added a phrase in the last part the sentence as "... except for the D2 domain of PrgH, which uses completely different surfaces for ring formation²²⁻²⁶".

8. line 206. where the hydrophobic residues are introduced, it might be worth noting explicitly that these side-chains, though hydrophobic, are surface-exposed on the individual subunits

(which I gather from the statement that they form the interfaces). How much hydrophobic contact area do they account for at the interface between subunits?

Re: It was not an accurate description that the subunit interface in the S ring is mainly mediated by the hydrophobic interactions. There are in fact many polar and hydrophobic residues involved in the subunit interface. We therefore rephrased the sentence as "The subunit interface of the RBM3 domains in the S ring is mediated by both the polar and hydrophobic interactions."

9. Line 227. Do these two mutants produce the same number of hook-basal bodies as the wild type, or is the number of HBBs diminished? If so, please note. If not, then it remains possible that they disfavor assembly of the MS ring-not enough to prevent assembly altogether, but enough to decrease the number of HBBs, and therefore, export apparatuses.

Re: Thank you for the question. Our description of the point may have been a little confusing. No, the number of HBBs the I252A and L253A mutants of FliF produced was significantly reduced compared to the wild-type but was not diminished. We therefore added a phrase "... but at a significantly lower level than the wild-type" to the sentence. But since we isolated HBBs with the filament attached first and then depolymerized the filament at a low pH to purify HBBs for negative stain EM observation (Fig. 3d lower panel), the reduction in the number of HBBs would also reflect the reduction in the level of protein export activity. The level of MS ring and C ring assembly was maintained as monitored by the level of FliF and FliN in the membrane fraction (Fig. 3c). Therefore, the reduction in the number of HBBs is due to an inefficient assembly of the export gate into the MS ring. We rearranged the order of relevant paragraphs and slightly modified the sentences to make it easier for readers to understand this.

10. line 240. Is it known whether localization of FliN requires the formation of normal C-rings? Is it possible that the mere presence of FliF, FliG, and FliM, whether they assemble into rings or not, can enable association of some FliN with the membrane?

Re: Yes, it is known that localization of FliN in the membrane fraction requires the C ring formation around the MS ring. In the absence of MS ring, even FliG and FliM are not associated with the membrane (Kubori *et al. J. Bacteriol.* 1997).

11. line 249. I was surprised to read here that two of the mutated positions are actually not at the interface between subunits. In the remarks introducing this mutational analysis section (lines 205-207), it seemed to imply that the aim of the mutations was to probe hydrophobic residues at the inter-subunit interface.

Re: We apologize for the confusing introductory remarks. The aim of the mutational analysis was to probe the effect of mutations of conserved residues. We therefore rephrased the introductory remarks as mentioned above.

11. If we grant that HBBs are made in normal numbers in the mutants, and that all of them contain normal C-rings, then it seems that assembly of the export apparatus into the mutant assemblies must be normal, because construction of the rod and hook depends on the action of the export apparatus. It might be argued that what's impaired is the export of the later subunits that form the filament; i.e., the FliF mutations are somehow selectively impeding export of the 'late' cargoes.

Re: As we wrote in response to your comment 9, the data shown in Fig. 3c indicate that the normal assembly of MS-C ring is maintained by these FliF mutant strains, but the number of HBBs with the filament was reduced by these mutations, indicating a significant reduction in the level of protein export activity. The reduction in the secretion level of export substrates was nearly the same for FigD, FigE and FliC as shown in Fig. 3e, and was not specific to the late substrates.

12. line 261. It is suggested that the crosslinked MS ring can no longer have the type III secretion

apparatus inserted. Was this (or could this be) tested by export assays like those described above for the other mutants?

Re: Yes, we analyzed the secretion levels of FlgD, FlgE and FliC by the I252C/H263C double mutant, which produces the crosslinked MS ring. As shown in the figure attached below, the I252C/H263C double mutation inhibited the secretion of FlgD, FlgE and FliC, thereby causing a complete non-motile phenotype (Fig. 3g,h).

13. line 306. The emphasis is on symmetry, which is certainly important, but I also wondered—given that the authors are saying that the PQR assembly with 5.5 subunits per turn is inserted into the central part of the MS ring that has 23-fold symmetry—whether the inner surface of the structure, which must mate to the PQR assembly, is smooth or rough. Does the 23-fold symmetry give rise to rough topography with 23-fold symmetry, or is it a fairly smooth structure, in which the symmetry is not really very important?

Re: As you pointed out, the inner surface of the 23-fold symmetry ring is quite smooth, and we do not think the 23-fold symmetry is really important here, either. It is the size of the hole with a smooth inner surface that would be important for the PQR assembly. To make this point clearer, we rephrased the sentence as “It is the internal core ring of the M ring formed by 23 subunits of RBM2 that forms a hole of the right size with a smooth inner surface to accommodate the export gate ...”.

14. line 314. Same point as above regarding ‘sort of’.

Re: We rephrased it as “This puzzle appeared to have been resolved ...”

15. line 353. This seems an unnecessarily harsh critique of the earlier study. Is there really any basis to say that all detailed features of the previous study were non-physiological?

Re: We apologize for the rather harsh comment on the earlier study. To make it milder, we rephrased the sentence as “... albeit the symmetry variation and some of the structural features may not be physiologically relevant, possibly due to the C-terminal truncations of FliF as confirmed by our present analysis (Supplementary Fig. 11).”

16. Fig. 2.a. This looks like a single averaged map, not a set of 2D class averages.

Re: This is a representative 2D class average image cut out from the result of 2D classification analysis. We rephrased the legend as “A representative 2D class average image showing 34-fold symmetry”.

REVIEWER COMMENTS

Reviewer #1 (Remarks to the Author):

I was excited to read the outcome of the comparison between full-length, 1-503, and 1-456 length variants of FliF, which nicely substantiate the reviewer's main claim (and pose more questions, albeit for future studies).

Reviewer #2 (Remarks to the Author):

The revised manuscript is greatly strengthened by including additional structures from two FliF fragments and by addressing some of previous concerns from three reviewers. However the data presented in the revised manuscript still seems insufficient to fully support the two major findings of the manuscript: 34-fold symmetry in the native MS-ring and 11-fold sub-symmetries. Specifically, 34-fold symmetry of the flagellar basal body should be clearly revealed at 6.8Å resolution. In contrast, it is very difficult to see the 34-fold symmetry in the class averages shown in Fig. 1 or Supplementary Fig. 1a, b.

Similarly, the 11-fold symmetry is poorly presented in Fig. 4d. Given that massive images were used for data analysis, a better structural information about different symmetries within the MS ring will be critical to distinguish this study from previous publications.

Specific suggestions:

1. In Fig. 1, the rod density appears to dominate the alignment and classification. To better show the symmetry of the native MS ring, it might be helpful to subtract rod density.
2. The 34-fold symmetry of the MS ring is not obvious in the supplementary fig. 6. It will be informative to show the density map as presented in the supplementary fig. 4.
3. It is difficult to see the 11-fold symmetry in fig. 4d. More importantly, the difference between fig. 4d and fig.4a or fig. 4b is so striking, suggesting that the 11-fold symmetry is less convincing.

Response to the reviewers' comments:

To Reviewer #1:

I was excited to read the outcome of the comparison between full-length, 1-503, and 1-456 length variants of FliF, which nicely substantiate the reviewer's main claim (and pose more questions, albeit for future studies).

Re: Thank you very much. We are glad you are satisfied with the revised manuscript including two new structures.

To Reviewer #2:

The revised manuscript is greatly strengthened by including additional structures from two FliF fragments and by addressing some of previous concerns from three reviewers.

Re: Thank you very much.

However the data presented in the revised manuscript still seems insufficient to fully support the two major findings of the manuscript: 34-fold symmetry in the native MS-ring and 11-fold sub-symmetries. Specifically, 34-fold symmetry of the flagellar basal body should be clearly revealed at 6.8Å resolution. In contrast, it is very difficult to see the 34-fold symmetry in the class averages shown in Fig. 1 or Supplementary Fig. 1a, b.

Similarly, the 11-fold symmetry is poorly presented in Fig. 4d. Given that massive images were used for data analysis, a better structural information about different symmetries within the MS ring will be critical to distinguish this study from previous publications.

Re: Thank you for your comments. We now improved the presentation of the relevant panels in the figures to make them more informative and convincing, as explained below.

Specific suggestions:

1. In Fig. 1, the rod density appears to dominate the alignment and classification. To better show the symmetry of the native MS ring, it might be helpful to subtract rod density.

Re: We improved the contrast of Fig. 1c as much as possible to show the 34-fold symmetry of the MS ring more clearly.

2. The 34-fold symmetry of the MS ring is not obvious in the supplementary fig. 6. It will be informative to show the density map as presented in the supplementary fig. 4.

Re: The 34-fold symmetry of the MS ring in the basal body was clearly presented in Fig. 4a of the original and revised manuscripts, but we added an end-on view of the 3D reconstruction to Supplementary Fig. 6 to show it there as well.

3. It is difficult to see the 11-fold symmetry in fig. 4d. More importantly, the difference between fig. 4d and fig.4a or fig. 4b is so striking, suggesting that the 11-fold symmetry is less convincing.

Re: We apologize for the difference in the scales of Fig. 4a,b and Fig. 4d. We magnified Fig. 4d to the same scale and put number labels from 1 to 11 for easier recognition of the 11-fold symmetry parts within the MS ring structure.

REVIEWERS' COMMENTS

Reviewer #2 (Remarks to the Author):

The previous concerns have been adequately addressed. This is a beautiful study with significant contribution to the flagellar field.